# Cancer and Stress: Does It Make a Difference to the Patient When These Two Challenges Collide?

**DOI:** 10.3390/cancers13020163

**Published:** 2021-01-06

**Authors:** Anem Iftikhar, Mohammad Islam, Simon Shepherd, Sarah Jones, Ian Ellis

**Affiliations:** Unit of Cell and Molecular Biology, The Dental School, University of Dundee, Dundee DD1 4HN, UK; AIftikhar@dundee.ac.uk (A.I.); m.r.y.islam@dundee.ac.uk (M.I.); S.D.Shepherd@dundee.ac.uk (S.S.); s.j.jones@dundee.ac.uk (S.J.)

**Keywords:** stress, HNC, glucocorticoid signalling, β-adrenergic signalling, cancer

## Abstract

**Simple Summary:**

Head and neck cancers are the sixth most common cancer in the world. The burden of the disease has remained challenging over recent years despite the advances in treatments of other malignancies. The very use of the word malignancy brings about a stress response in almost all adult patients. Being told you have a tumour is not a word anyone wants to hear. We have embarked on a study which will investigate the effect of stress pathways on head and neck cancer patients and which signalling pathways may be involved. In the future, this will allow clinicians to better manage patients with head and neck cancer and reduce the patients’ stress so that this does not add to their tumour burden.

**Abstract:**

A single head and neck Cancer (HNC) is a globally growing challenge associated with significant morbidity and mortality. The diagnosis itself can affect the patients profoundly let alone the complex and disfiguring treatment. The highly important functions of structures of the head and neck such as mastication, speech, aesthetics, identity and social interactions make a cancer diagnosis in this region even more psychologically traumatic. The emotional distress engendered as a result of functional and social disruption is certain to negatively affect health-related quality of life (HRQoL). The key biological responses to stressful events are moderated through the combined action of two systems, the hypothalamus–pituitary–adrenal axis (HPA) which releases glucocorticoids and the sympathetic nervous system (SNS) which releases catecholamines. In acute stress, these hormones help the body to regain homeostasis; however, in chronic stress their increased levels and activation of their receptors may aid in the progression of cancer. Despite ample evidence on the existence of stress in patients diagnosed with HNC, studies looking at the effect of stress on the progression of disease are scarce, compared to other cancers. This review summarises the challenges associated with HNC that make it stressful and describes how stress signalling aids in the progression of cancer. Growing evidence on the relationship between stress and HNC makes it paramount to focus future research towards a better understanding of stress and its effect on head and neck cancer.

## 1. Introduction

Head and neck cancers (HNC) are a globally growing burden resulting in significant mortality and morbidity [1]. With around 650,000 cases diagnosed every year, HNC are the sixth most common form of cancers in the world [2]. More than 90% of the head and neck cancers are squamous cell carcinoma (SCC) of the oral, laryngeal or oropharyngeal mucosal surfaces [3]. More than 90% of the malignant tumours of the oral cavity are SCC, with significantly fewer neoplasms presenting in the soft tissues and minor salivary glands [4].

HNC are stressful for patients and their caregivers from diagnosis, throughout the course of treatment and persist in survivorship [5,6,7]. The involvement of surgery, radiation and/or chemotherapy makes the treatment course even more complex and tedious for the patients. The overall impact is momentous, as patients struggle with the myriad of challenges, all of which act as precursors to stress [8] (Figure 1). This review highlights the challenges that become a source of stress and reduced quality of life (QoL) in HNC patients. It also inspects the response of the body to stress and how the stress pathways, as studied in other types of cancers, aid in tumour progression. Despite HNC being amongst the most stressful cancers, the studies on its role and effect in tumour progression are still scarce and are summarised in Table 1.

Amongst the many challenges faced by HNC patients, the diagnosis itself is considered a serious problem, as it changes their perspective on life. One of the first works in the literature that identified cancer as an intense emotional distress dates back to 1951 [22]. Psychiatrists have observed reactions of patients with breast, ovarian, cervical cancers and tongue lesions. For most patients, cancer was an intense emotional distress and meant a painful and lingering death. The patients described the news of cancer “a heavy blow to the head”, as they looked towards the future with fearful apprehensions [23].

The head and neck region has an enormous cosmetic importance. It plays a central role in social interactions and identity [24]. Increasing tumour size or treatment modalities may result in disfigurement. A study suggests 41% to 71% of patients diagnosed with HNC require surgical intervention [25]. Changes in facial appearance due to treatment or disease can have devastating consequences for the patients, especially since the deformity is highly visible compared with other types of cancer [26]. Patients suffer from negative emotions and a distorted view about self [27,28,29,30]. This can also compromise the functional aspects of eating, speech, breathing and body image [31,32,33,34,35,36]. Additionally, the stigma attached to facial disfigurement may impact upon social and family interactions as well as work life, leading to self-esteem issues [37,38,39,40]. As a consequence of these difficulties patients often experience depression, social anxiety and a generalised sense of reduced quality of life (QoL) [27,32,41,42].

Studies have used different screening tools to assess the relationship between QoL and disfigurement. A systematic review by Djan in 2013 [43], considered five QoL questionnaires suitable for HNC patients. These included, University of Washington QoL (UWQOLQ), Head and Neck Survey (HNS), Europe Organisation for Restoration and Treatment of Cancer QoL Questionnaire Head and Neck 35 (EORTC-QOQ-H&N35), Derriford Appearance Scale 24 (DAS24) (DAS 59). The UWQOLQ was found to be most appropriate for appearance issues in clinical practice, whereas DAS24 and DAS 59 were important screening tools in understanding the effect of appearance on QoL in a research setup. The extent of facial disfigurement was reported to be negatively associated with psychological and social function [44,45]. Wang et al., in 2018, used questionnaires including a Facial Disfigurement Scale, Social Support Scale and Psychosocial Adjustment to Illness scale and reported poor psychosocial adjustment in Oral cancer patients with severe disfigurement [46]. The study [47] employed a Functional Assessment of Cancer Therapy-Head and Neck (FACT H&N) questionnaire and observed that disfigurement was significantly associated with the functional dimension of patients’ QoL. It was also reported that disfigurement was positively associated with psychological distress when social self-efficacy was low [48].

After a diagnosis of cancer, the concern or fear that cancer may return or progress, is also seen in patients. This is termed as the fear of recurrence (FCR) [49,50,51]. Early studies reported that FCR was a prominent apprehension of all the cancer patients [52]. In patients diagnosed with breast, prostate and lung cancer, FCR was recognised as the most commonly reported worry [53]. FCR is a frequent concern for HNC patients and is associated with psychological stress [54,55]. The significant positive relationship between psychological distress and FCR remained stable at the two time points of 3 and 6 months post diagnosis [55].

Follow up appointments and somatic symptoms triggered FCR [56,57]. FCR was elevated by deterioration of somatic symptoms and was shown to be a link between somatic symptoms and stress [58]. FCR was also reported to be a mediator between severity of symptoms and QoL in HNC patients. The specific symptom of pain was significantly related to FCR [59]. Prevalence of FCR was variously reported to affect 31% to 61% patients with HNC [60,61]. A recent prospective study reported a high FCR in 52.8% of HNC patients. Higher levels of anxiety, younger age, introversion and previous smoking habit were significantly associated with FCR [62].

Cancer patients commonly suffer from pain. It is one of the most frightening symptoms of cancer [63]. The International Association for the Study of Pain defines it as “an unpleasant sensory and emotional experience with actual or potential tissue damage” [64]. The studies in HNC have observed the prevalence of pain over different time points from diagnosis up to the treatment completion and follow-up. Gellrich et al., 2002, reported that 56% of HNC patients presented with pain at the time of diagnosis and 96% of these patients had mixed neuropathic and nociceptive pain, whereas Potter et al., 2003, reported pain as a less common symptom at diagnosis [65,66]. In another study by Ribeiro et al., 2003, 60% of patients with oral and oropharyngeal cancers reported pain [67]. The rich nerve supply and the presence of vital anatomical structures in the confined space of the head and neck region make it extremely sensitive to pain [68]. HNC patients frequently complain of pain [69,70]. A meta-analysis by Everdingen et al., in 2007, reported the prevalence of pain in head and neck cancers to be higher compared with gastrointestinal, lung and breast cancers [71]. The functional role of this region increases the significance of pain for the patients [63]. In a sample of 113 oral squamous cell carcinoma patients, 37% reported spontaneous pain and 68% reported function-related pain [72]. A study by Breivik et al., 2009, showed 86% patients with head and neck squamous cell carcinoma (HNSCC), reported pain [69]. Studies have shown the direct association of pain with poor quality of life [73,74,75,76,77].

HNC is a debilitating illness with disease challenges and complex treatment plans. Consequently, patients have to endure a profound emotional impact due to the diagnosis itself, demanding treatment plans, compromised function, aesthetics and disrupted social life. The anatomical importance and complexity of HNC requires a multi-faceted rehabilitation, a part of which must also address the psychological stress of these patients to improve the overall success of treatment [6,7].

## 2. Response to Stress

The challenges/stressors associated with the diagnosis and treatment of cancer engender emotional distress and negatively affect health-related quality of Life (HRQoL) [78]. In addition, HNC patients with depressive symptoms showed lower overall survival, less motivation and poor response to treatment [79]. Evidence suggests that there is a role of neuro-endocrinological markers between HRQoL, psychological stress and cancer survival [80,81,82]. The key biological responses to stressful events occur through the combined action of two systems, i.e., the hypothalamus–pituitary–adrenal axis (HPA) and sympathetic nervous system (SNS), both activated by the central nervous system (CNS), (Figure 2) [83].

Any stressful situation is recognised by the paraventricular nucleus of the hypothalamus, that plays a role of a biological circuit, integrating human experiences with physiological signalling and releasing the corticotropin-releasing hormone (CRH) [84]. CRH acts on the pituitary gland, which then releases adrenocorticotropic hormone (ACTH) (Figure 2). ACTH signals the adrenal cortex to release glucocorticoids. The glucocorticoids, mainly cortisol, are released in an inactive form, cortisone and converted to the active form cortisol and back to cortisone by the enzymes, 11-β hydroxysteroid dehydrogenase type 1 (11-β HSD-1) and 11-β hydroxysteroid dehydrogenase type 2 (11-β HSD-2), respectively, in the target organs (Figure 2). They increase lipolysis and gluconeogenesis to increase the available energy sources. A negative feedback system regulates production of cortisol via the hypothalamus and pituitary gland [85].

The SNS, on the other hand, is activated by the autonomic nervous system (ANS) (Figure 2). Once activated, it stimulates the adrenal medulla to release the catecholamines, epinephrine and norepinephrine (Figure 2). The catecholamines exert effects on cardiovascular, pulmonary, hepatic, skeletal and immune systems for quick transport of energy to the organs [83]. As a result, homeostasis is re-established, provided the stressor falls into the adaptive capacity [84].

Signals from the HPA and SNS shape the stress response of the body and enable it to survive through the stressful event. However, prolonged exposure to stress can result in the body being negatively affected by the stress hormones, a phenomenon called allostatic load whereby the demand exceeds the adaptive capacity of an individual [86]. The stress response may then be accompanied by changes in the defence mechanisms, metabolism and circulation, resulting in high blood pressure, altered immunity and cytokine levels [83]. Activation of these systems over a long period of time increases the levels of glucocorticoids and catecholamines [87,88,89]. Studies have shown that chronic stress, leading to increased levels of catecholamines and glucocorticoids, plays a role in cancer progression [90].

### 2.1. Adrenergic Signalling Pathway

The adrenergic pathway mediates the sympathetic nervous system-induced fight and flight stress responses. It functions through adrenergic receptors and the neurotransmitters—epinephrine (adrenaline) and norepinephrine (noradrenaline) [91]. The receptors involved include α-adrenergic and β-adrenergic receptors [92]. These receptors belong to a family of G-protein-coupled receptors, which consist of seven transmembrane spanning domains, three intracellular and three extracellular loops, one extracellular N-terminal domain and one intracellular C-terminal domain. The α-adrenergic receptors show affinity for norepinephrine (noradrenaline), while the β-adrenergic receptor shows affinity for epinephrine (adrenaline) [93]. The sub-types of β-adrenergic receptors are β1, β2, β3. These three types of β-adrenergic receptors are found on multiple sites of tumour growth and metastasis, such as brain, liver, lungs, breast, ovary, prostate, lymphoid, bone marrow and vasculature [91,94].

The interaction of catecholamines with their receptors can activate multiple signal transduction pathways involved in survival and apoptosis. Even in non-stressed states, they are involved in the regulation of blood pressure and heart rate [93]. The binding of epinephrine (adrenaline) and norepinephrine (noradrenaline) to β-adrenergic receptors results in the activation of G guanine nucleotide binding protein leading to the stimulation of adenylyl cyclase synthesis of cyclic AMP [94], (Figure 3). The cyclic AMP, in turn, is involved in the regulation of multiple cellular processes through Protein Kinase A (PKA) and Exchange Protein Directly Activated by cyclic AMP (EPAC)—the two major effector systems (Figure 3), [91,95,96].

PKA, activated by cyclic AMP, results in the phosphorylation of transcription factors (Figure 3) [94,97,98,99]. PKA is involved in the regulation of various cellular processes such as growth, metabolism, differentiation, morphology, neurotransmission and gene transcription [91]. It also phosphorylates β-arrestin receptor kinase (BARK), which results in β-arrestin inhibition of β-adrenergic signalling and activation of Src (Figure 3) [99]. Src activates Focal Adhesion Kinase (FAK) which increases cytoskeletal rearrangements and cell motility [94]. Cyclic AMP (cAMP)-PKA activated by stress hormones also leads to increased proliferation and angiogenesis via PI3K/AKT/mTOR/P70S6K/HIF1α pathway. In cervical cancer cells, PKA dephosphorylates Yes-Associated Protein (YAP). The translocation of dephosphorylated YAP into the nucleus leads to inhibition of apoptosis [100], (Figure 3).

The second major effector is EPAC, (Figure 3) [95]. EPAC activates RAS (Rat Sarcoma)-like guanine triphosphatase RAP1A, which in turn stimulates B-RAF, MAP/ERK1/2 and ERK1/2 with effects on cell growth and proliferation [96], while PKA predominantly exerts effects on inflammation, angiogenesis and invasion, EPAC results in changes in cell morphology and motility [94].

### 2.2. Glucocorticoid Receptor Signalling

Our everyday functioning is dependent on several physiological processes. These processes such as metabolism [101], immune defence [102], growth and development [103], mood stability [104,105], essentially all involve the role of GC.

A brief look at the historical journey of GC, highlights the names of Han Selye and Phillip Hench. Selye focused on glucocorticoids from a stress point of view and was the first to discover that cortisol released from the adrenal cortex participated in the stress response. He also demonstrated that glucocorticoids exert strong anti-inflammatory effects [106]. At the same time, Hench, while treating his Rheumatoid Arthritis (RA) patients, observed that a substance X appeared, during jaundice and pregnancy, that resulted in a cure of RA. Hench and Kendle later discovered that this substance was cortisol and received a Nobel prize for isolating and identifying the structure of glucocorticoids [107]. Since then, the synthetic form of cortisol, dexamethasone has been widely celebrated as a wonder drug in a wide array of diseases. However, the deregulated levels of cortisol in the body, as well as the exogenous dexamethasone, act as a double-edged sword in cancer, suppressing some forms and progressing others.

Glucocorticoids mediate their actions through the glucocorticoid receptor (GR), a ligand-inducible transcription factor [108,109]. The glucocorticoid receptors are the fundamental directors in events that follow stress exposure. They are crucial for the stress response, as well as in the treatment of autoimmune, inflammatory and allergic diseases [110].

The human glucocorticoid receptor (hGR) gene called NR3C1 is found on chromosome 5q31.32 (Figure 4). It consists of nine exons and spans 160 kbs. Alternative splicing of the hGR gene in exon 9, results in two receptor isoforms GRα and β, (Figure 5) which differ at the ends of their C-termini [111]. GRα contains 50 additional amino acids in its ligand-binding domain (LBD) with a molecular weight of 97 kDa and GRβ containing 15 amino acids with a molecular weight of 94 kDa, (Figure 5) [112]. GRα-specific sequences enable its binding to glucocorticoids and the recruitment of coregulator by AF-2 [111]. GRβ does not bind the glucocorticoids, however it can bind the receptor antagonist, mifepristone. Grβ is known to be a dominant-negative regulator of GRα. Genome-wide analysis showed that hGRβ can alter the activity of genes controlled by GRα [113]. GRs are expressed in malignancies and the intensity of their expression differs according to the tissue.

The human GR (hGR) is composed of three major domains, each with a specific function. The N-terminal domain (NTD), composed of the first 421 amino acids, possesses a major transactivation domain termed, Activation Factor-1 (AF-1). AF-1 which is a transcription factor plays a critical role in the regulationof the receptor by acting as a site of communication between the NTD and coactivators, chromatin modulators and basal transcription factors, such as RNA polymerase II and TATA binding protein [114,115].

The DNA-binding domain (DBD) consists of amino acid residues from 420 to 480. This domain is essential for the binding of GRα to the specific DNA sequences, Glucocorticoid Receptor Elements (GRE), through the two C4-type zinc fingers [116,117]. The DBD also contains the amino acids for GR homo-dimerisation [118]. In addition, it also contains the nuclear localisation domain (NLS1) [119,120]. The ligand-binding domain (LBD) of the hGRα consists of amino acid residues 481 to 777. This domain is comprised of 12 helices that contain a hydrophobic pocket and form a ligand-binding site, which binds the glucocorticoids and therefore plays a critical role in the ligand-induced activation of GRα [121]. Ligand binding results in a conformational change through these helices in the LBD. The new conformation closes the ligand-binding pocket and enables GRα to interact with importin in the nucleus, members of the transcription initiation complexes and transcription factors that are essential for the ligand-dependent activation of GRα [109]. Another nuclear localisation domain (NLS 2) is also found on the LBD and acts only as a weak localisation signal [119].

Prior to ligand binding, GR is found in the cytoplasm bound to a GR chaperone complex (Figure 6). The chaperone complex consists of the heat shock proteins (Hsp 70, Hsp 90, Hsp 40) and p23 immunophilin, which maintain the inactive form of GR and enable the ligand-binding domain to identify the ligand and block the nuclear localisation sequence (NLS), inhibiting its translocation into the nucleus [122]. HSP 40 and immunophilin p23, also play an essential role in the association of different proteins in the complex and GR maturation [123]. GR goes through post-translational modifications such as phosphorylation, acetylation, SUMOylation and ubiquitination that effect its action [124].

Upon classical ligand binding, the NLS is exposed and the GC-GR complex translocates into the nuclear pore and the accessory proteins are transferred back into the cytoplasm via importin [126]. Studies in breast cancer show ligand–receptor binding causes a conformational change in the structure and GRα is phosphorylated at serine 211 by p38 MAPK. This leads to dissociation of the inhibitory proteins and exposes the NLS and dimerisation domain (DD). GR alpha dimerises and translocates into the nucleus via the nuclear pore [127]. Ten serine residues have been identified as phosphorylation targets. These include S45, S113, S141, S203, S211, S226, S236, S267, S404, S134 and T8 and some of them are associated with inhibition of GC signalling [112,128]. A recent study on Triple-Negative Breast Cancer (TNBC) also showed that TGFβ promoted ligand-independent, p38 MAPK—induced S134 GR phosphorylation, which resulted in migration and invasion [129].

Translocation of the GC-GR complex in the nucleus generates genomic effects resulting in either activation or repression of gene transcription [109]. Activation or repression can occur through direct DNA binding or through indirect DNA binding (Figure 6) [130]. The ligand-bound GR can directly bind to the positive GRE (GRE) or negative GRE (nGRE) to bring about activation or repression of transcription, respectively [130]. The binding of the GC-GR complex to GRE, followed by GRE dimerisation, leads to recruitment of cofactors, such as CBP, P300 and histone acetyl transferases (HAT), resulting in gene expression [110]. The binding of GR to nGRE prevents the interaction of transcription factors with promoter DNA and results in gene silencing. Prolactin, neuronal serotonin receptor, corticoid-releasing hormone and vasoactive intestinal peptide are examples of genes that contain nGRE.

Few inflammatory genes repressed by GC are identified to contain these GREs [131]. It is the concentration of GC that governs which GREs will be occupied by the GC-GR complex. The tissue-specific effects of GC could be the result of chromatin accessibility and GRE’s distinct sensitivity [132]. Gene activation and repression by GR can also take place independently of direct DNA binding, through interaction with other transcription factors such as NFk B, TGF β, MAPK and STAT [127]. GR location, protein–protein interaction (PPI) and sensitivity to GC, affect the GC efficacy. GC-GR effects are dependent on tissue and cell type and may also differ according to the host condition [110,119].

Increasing evidence indicates activity through non-genomic signalling mechanisms as well, resulting in rapid cellular responses that occur over a few seconds to minutes (Figure 6). These do not involve changes in gene expression, but activation of signal transduction pathways. These mechanisms occur via membrane-bound GR (mGR) and cytoplasmic GR [133,134].

Membrane-bound GR is reported to activate p42 MAPK signalling protein [135]. In skin cancer, it inhibits the PI3K-AKT pathway [136], whereas in the heart, GR activates the PI3K-AKT pathway. In the last decade, research studies have looked at the effect and the mechanism by which this receptor and its ligand, cortisol, play a role in cancer. Where certain cancers describe cortisol as a culprit of metastasis and resistance to treatment, others are suggestive of its suppressor properties in the disease [137].

## 3. Stress Hormones (Glucocorticoids and Catecholamines) and Their Effect on the Biology of Cancer

Cancer development is a multistep process including initiation, promotion and progression, where due to oncogenic mutations, normal cells become malignant [138,139]. Multiple signalling pathways are involved in cell growth, expansion and proliferation. Oncogenic mutations disturb the normal functioning of these signalling pathways, leading to abnormally growing cells and their resistance to apoptosis [140,141].

The progression of cells from normal to a neoplastic state requires acquisition of traits collectively termed “hallmarks of cancer”. These include, proliferative signalling, evasion of growth suppressors, resistance to cell death, replicative immortality, angiogenesis, invasion and metastasis [140]. These characteristics are not acquired by the tumour cells alone, rather they are the outcome of interaction between the tumour cells and the normal cells of the host which constitute the tumour microenvironment [142].

Research has highlighted the role of catecholamines and glucocorticoids in almost every facet of the multistep process of cancer [143]. However, studies on the effect of stress in head and neck cancer are still scarce in comparison to other cancers and these are presented in Table 1. Recent studies have explored that resilience in head and neck cancer patients may help alleviate the symptoms of anxiety and it was positively associated with hope and optimism [144].

### 3.1. Role of Stress Hormones in Cell Proliferation

Cells proliferate as a result of signals from various signalling pathways and their growth is dependent on the net balance between positive and negative signals [142]. Studies have shown increased levels of circulating catecholamines and glucocorticoids in response to stress, resulting in pro-tumuorigenic activities [11,145]. A study reported increased proliferation of gastric cancer cells in response to epinephrine and norepinephrine at 10 µM, however the proliferation effects were insignificant at higher doses. It was also observed that stressed mice showed remarkably high tumour weight and reduced physical activity. The effect on proliferation was blocked by the β-adrenergic blocker, propranolol. Propranolol also blocked the tumour cell growth by arresting cells in the G1/S phase and also inhibited ERK1/2-JNK-MAPK pathway, suggesting its role in growth and proliferation of gastric cancer cells. The expression of β2-adrenergic receptors on gastric cancer tissues was higher than normal samples and activation of these receptors was related to increased malignancy of gastric cancer [146]. Nicotine also interacts with β2-adrenergic receptors and activates the downstream COX 2 pathway, which can cause cell proliferation in gastric carcinoma and colon cancer [147,148]. Studies in breast cancer have also shown that catecholamines can cause increased tumour growth and proliferation [149,150]. Overexpression of β-adrenergic receptors has also been reported in breast cancer [151]. Patients with stage 1 breast cancer showed a significant reduction in Ki-67-based breast tumour proliferation in response to β blockers [152]. Proliferative effects of adrenaline were also reported in colon cancer [153]. Epinephrine and norepinephrine increased proliferation in vitro, as well as increasing tumour growth in vivo through activation of ERK1/2 by adrenergic signalling [154]. Lung tumour cell proliferation was enhanced by β-AR agonist isoproterenol and blocked by propranolol [96]. Studies on oral squamous cell carcinoma (OSCC) cell lines showed norepinephrine increased the proliferation of cells. OSCC cells and biopsies expressed β1- and β2-adrenergic receptors [9]. An oesophageal squamous cell carcinoma cell line also showed the expression of β1- and β2-adrenergic receptors. Stimulation by epinephrine resulted in cell proliferation. It also increased ERK ½ phosphorylation and expression of COX-2, Cyclin D1, Cyclin E2, CDK 4, CDK 6, VEGF and VEGF receptor in a β-adrenergic, MAPK/ERK and COX 2-dependent manner [155]. Higher β2 receptor expression was related to lymph node metastasis [13]. Another study showed that a strong β2-adrenergic receptor expression in oral cancer was linked to a higher overall survival [12]. β3-adrenergic receptors found in high concentrations in melanoma were also found to be crucial in proliferation of melanoma cells via nitric oxide signalling [156]. In contrast, studies also show the inhibitory effect of the β2-adrenergic agonist on breast cancer by blocking the Raf1/Mek1/Erk1/2 pathway [157,158]. This suggests that the effect of stress hormones may vary according to the tumour type, cell line, receptor expression and selectivity of the β-blocker [159].

The role of cortisol in tumour cell proliferation has been observed in many cancer types. Cortisol induced a two-fold proliferation in human mammary carcinoma cell lines, suggesting that cortisol concentrations in the body may cause growth of breast cancer [160]. In contrast, an anti-proliferative effect of dexamethasone was observed in the MCF-7 cell line [161] and the 3AO human ovarian cancer cell line [162].

Dexamethasone enhanced proliferation and metabolic activity in the cell lines from lung carcinoma (CALU-6), mammary carcinoma (MCF-7, MDA MB-231), Ewing’s sarcoma (CADO), rhabdomyosarcoma (RH30), glioblastoma (A172, U118), neuroblastoma (SHEP). Addition of mifepristone (RU486) blocked dexamethasone-induced proliferation. Cell lines that did not express GR, did not undergo proliferation. Dexamethasone also resulted in phosphorylation of AKT and P38 MAPK in the MCF-7 cell line, as well as their downstream effectors GSK-3 and HSP 27. Inhibition of AKT and MAPK blocked the dexamethasone-induced proliferation [163]. Dexamethasone-activated GR induced the transcription of Serum/Glucocorticoid Regulated Kinase 1 (SGK) in a rat mammary tumour cell line. SGK protected the cells against apoptosis [164]. SGK plays a role in proliferation, migration, growth and survival and is downstream of PI3K signalling [165]. Studies have shown the role of SGK in tumour development and cell survival [166,167]. SGK was found to be over-expressed in nasopharyngeal carcinoma and blocking SGK suppressed cell proliferation and migration [168]. A study showed upregulated SGK, as a result of increased glucocorticoids, increased the expression of MDM2 which in turn decreased p53 [169]. On the contrary, annexin-1, a glucocorticoid-induced protein with inhibitory effects on cell proliferation showed reduced expression in nasopharyngeal carcinoma in comparison to that in normal tissue [170].

### 3.2. Role of Stress Hormones in Angiogenesis

Angiogenesis is a characteristic hallmark of cancer and an important feature in its progression [140]. In pathological disturbances, such as cancer, the angiogenic signalling pathways remain switched on, thus aiding tumour growth [171]. The inducers of angiogenesis include members of the vascular endothelial growth factor (VEGF) family, angiopoietins, transforming growth factors (TGF), platelet-derived growth factors (PDGF), tumour necrosis factor alpha (TNF-α), interleukins and members of the fibroblast growth factor family (FGF), insulin-like growth factor and hypoxia inducing growth factor (HIF1), [94,171,172]. Tumours from stressed animals showed increased expression of VEGF, matrix metalloproteinase (MMP)2 and MMP9. β-adrenergic activation of cAMP-PKA resulted in increased tumour growth and angiogenesis [146]. Surgical stress also increased micro vessel density and VEGF expression, which was blocked by propranolol [173]. Clinical studies showed decreased serum levels of VEGF correlated with increased social support in ovarian carcinoma patients [174]. Decreased levels of IL-6 also correlated with increased social support [175]. β-adrenergic signalling also upregulated VEGF expression in human prostate cancer cell lines [176]. Another study reported adrenaline and noradrenaline-mediated inhibition of apoptosis in prostate cancer models but found no increase in angiogenesis [177]. However, angiogenesis was seen in different prostate cancer models [178], which showed that stress-activated β-adrenergic signalling and its downstream effector CREB, promoted histone deacetylases (HDAC), which inhibited the expression of TSP1 (an inhibitor of angiogenesis). Chronic stress increased expression of VEGF, MMP-2, MMP-9, in animal models of lung cancer [179]. Activation of β2 receptors on endothelial cells and tumour-associated macrophages (TAM), increased VEGF and angiogenesis, which was inhibited by the use of propranolol [96,180,181]. β3 receptor expression in human melanoma increased tumour aggressiveness, angiogenesis and promoted malignancy [182]. Blocking β3 adrenoreceptor in neuroblastoma cells led to reduction in tumour growth [183]. Epinephrine and norepinephrine also upregulated the expression of IL-6, VEGF, MMP-2 and MMP-9 in oral cancer cell lines, mouse models, nasopharyngeal call lines and human melanoma cell lines [9,10,184,185]. Norepinephrine and epinephrine also directly activate STAT, which promotes angiogenesis, cell survival and proliferation [186].

A study found deregulated gene expression in adrenergic and GC stress pathways in men with lethal and non-lethal tumours of the prostate. The glucocorticoid signalling pathway showed a strong association with lethal tumours. It was also associated with cell proliferation, angiogenesis and perineural invasion [187]. Other studies have also shown suppression of angiogenic factors in response to GC treatment [188]. Cortisol-activated GR in mouse mammary tumour cells caused DNA damage and increased the levels of reactive nitrogen species (RNS). Cortisol also upregulated VEGF and TWIST 1 expression, promoting angiogenesis and invasiveness [189]. On the contrary, studies have also reported that GCs downregulate angiogenesis in prostate cancer [188], bladder cancer [190], glioblastoma [191], melanoma [192], lung cancer [193,194]. In a study, dexamethasone downregulated the expression of MMP, inhibiting the invasive characteristics of a pancreatic cancer cell line [195]. In pancreatic ductal adenocarcinoma cell lines, dexamethasone concentrations, similar to those found in plasma-induced epithelial–mesenchymal transition (EMT), activated RAS/JNK signalling, increased TGF beta, vimentin and SOX-2 expression [196]. Dexamethasone also promoted tumour progression of melanoma cells in mouse models by increasing the activity of Rho-associated kinases (Rock1/2). Inhibiting Rock1/2 blocked the metastatic effects of GC [197].

### 3.3. Role of Stress Hormones in Invasion and Migration

Metastasis is a vital step in cancer progression and a leading cause of death [198]. Invasion and metastasis involve a series of cell biological changes that include local invasion, intravasation by cancer cells into nearby blood and lymphatic vessels, escape of cells into distant tissues (extravasation), formation of small nodules (micro-metastasis) and their growth into macroscopic tumours [140]. Studies have highlighted the impact of β-adrenergic signalling on invasion and metastasis [199,200,201,202,203,204] and its blockade by use of beta blockers [9,96,205,206]. A recent study showed β-adrenergic receptor-driven metastasis in breast cancer cell lines. Norepinephrine treatment increased the invasive ability of cells, which also showed the upregulation of the pro-metastatic gene, LYPD3 [207]. Another study showed stress hormones decreased the deformability of breast cancer cells, making them stiffer and more invasive [208]. The increased expression of MMP2 and MMP9 due to β-adrenergic signalling, also increased tissue invasion [10,184,186,204]. In OSCC, the expression of β2-adrenergic receptors was correlated with cervical lymph node metastasis and norepinephrine-induced migration of cell lines [19]. Stress hormones also induced EMT in lung cancer [209], ovarian cancer [210], prostate cancer [211] and gastric cancer [212].

### 3.4. Role of Stress Hormones in Cell Survival

Evasion of cell death processes such as apoptosis, necrosis and autophagy are essential for the progression of malignant tumours and for metastatic spread [140]. Ovarian cancer cells showed resistance to anoikis, when exposed to epinephrine or norepinephrine [213]. In ovarian cancer patients, stress behaviour was related to higher levels of phosphorylated FAK Y397, which correlated with high mortality. Another anti-apoptotic mechanism of epinephrine involved cAMP-dependent phosphorylation and inactivation of pro-apoptotic protein BAD. Upon phosphorylation, BAD releases Bcl-2 and B-cell lymphoma-extra-large (Bcl-xL), thereby inhibiting the apoptotic process [214]. A study on mouse models of prostate cancer showed that stress hormones inhibited apoptosis in PI3K inhibitor-treated mice. These effects were mediated by the ADRB/PKA/BAD anti-apoptotic signalling pathway [179]. Norepinephrine induced the β-adrenergic/PKA activation of YAP-1, regulating the Hippo-YAP1 pathway resulting in anoikis resistance and tumour progression in cervical cancer cells (Figure 3) [100].

Cortisol also stimulated the growth of prostate cancer cells by activating the androgen receptor in the absence of androgens [215]. The stress hormone pathway induced the proliferation of metastatic colon cancer cells by upregulating the expression of CDK1 [216]. CDK1 promotes G1-S transition. A dysregulation of CDK is associated with proliferation activity in tumours [141]. Several mechanisms have been identified behind the pro-apoptotic actions of glucocorticoids. Both endogenous and exogenous glucocorticoids had anti-apoptotic effects on human ovarian carcinoma cells lines by up regulation of the anti-apoptotic protein CIAP 2 and CL100 [217]. Glucocorticoids induced the expression of ciAP2 in lung cancer cells, leukaemia T cells and glioblastoma cells [218]. The anti-apoptotic protein Bcl-x was upregulated by glucocorticoids in fibrosarcoma [219]. Astrocytoma cells were also protected from apoptosis by glucocorticoid-mediated upregulation of Bcl-x anti-apoptotic protein [220]. In a study on human cervix and lung carcinoma, dexamethasone down-regulated cisplatin-induced expression of cell death receptor pathway components such as CD95-L, TRAIL, FADD and CASPASE-8 in carcinomas, as opposed to the pro-apoptotic actions in lymphoid carcinomas [221]. The apoptotic signalling is GR mediated, since the use of RU486, blocked the anti-apoptotic actions of DEX. Glucocorticoids also activated the PI3K/AKT pathway causing inactivation of pro-apoptotic molecules, for example, caspases [222]. Cortisol concentrations that depict physiological levels of 10nM caused increased secretion of IL-6 and proliferation of OSCC cell lines [9]. Dexamethasone increased the resistance of hepatocellular and colorectal tumours to cytotoxic therapy [223]. Dexamethasone also enhanced the growth of breast cancer, melanoma, neuroblastoma and cervical cancer cell lines treated with 5-Fluorouracil or cisplatin [224]. A recent study showed GC increased the GR-dependent expression of TEA Domain Transcription Factor 4 (TEAD 4). High TEAD 4 expression correlated with cancer cell survival, metastasis and poor survival of breast cancer patients [225]. In thyroid cancer cells, dexamethasone blocked TRAIL-induced apoptosis and upregulated the anti-apoptotic protein Bcl-xl, protecting the cancer cells. The use of mifepristone reversed this effect, suggesting the apoptosis blockade is mediated by the glucocorticoid receptor [226]. A decreased urinary concentration of glucocorticoids was found in thyroid cancer patients in comparison to normal controls [227].

### 3.5. Role of Stress Hormones in DNA Damage

Studies have shown the role of stress hormones in DNA damage and altered repair by production of reactive oxygen species (ROS) and reactive nitrogen species (RNS) [228,229]. A study on breast cancer showed that norepinephrine or cortisol increased the levels of ROS and RNS resulting in DNA damage. This was blocked by antagonists. An increase in inducible nitric oxide synthase (iNOS), was also observed in cortisol-treated cells. Cortisol-induced iNOS and DNA damage was reduced by using iNOS inhibitors [189,230]. DNA methylation has been linked to radiation resistance. A HNSCC radiation-resistant cell line, rSCC, showed a significant increase in DNA methylation compared to a radiation-sensitive cell line. Ingenuity pathway analysis in the same study showed glucocorticoid signalling as one of the top canonical pathways associated with radiation resistance [231].

Studies have also investigated the expression of GR in different tumours. In breast cancer tissue over-expression of GR in malignant epithelium compared to normal and lactational epithelium was found [232]. TNBC tumours expressed elevated expression of GR in 40% tumours [233]. Increased expression of the glucocorticoid receptor is associated with resistance to chemotherapy, metastasis [234] and shortened disease-free survival [235]. In another study, GR stromal expression directly correlated with tumour grade [236]. In ovarian cancer, high expression of GR was found in high grade and advanced stage tumours [237] and was associated with decreased survival in these patients. This was independent of BRCA status [238]. In prostate cancer, GR expression was reduced in primary prostate cancer, but restored in metastatic cancer [239]. On the contrary, increased nuclear expression of GR was linked to good prognosis and small tumour size, in a study on breast cancer [240]. In prostate cancer, GR expression was decreased in tumours. In cell lines, GC decreased proliferation, upregulated p21, p27, downregulated cyclin D1 and C-Myc phosphorylation [241]. In a recent study on the expression of GR in salivary duct carcinoma (SDC), a high GR expression was significantly associated with low five-year survival. A GR antagonist, mifepristone, decreased cell proliferation and cell survival in GR over-expressing cells [242]. Studies have shown tumour suppressor effects of GR in skin [243,244]. In a mouse model of skin cancer, increased expression of GR, decreased Akt activity and acted as a tumour suppressor by interference with NFkβ [136,245]. In non-melanoma skin cancer, the enzyme 11β HSD-2 which converts the active form of cortisol to the inactive form, cortisone, was seen to be upregulated and inhibition of this enzyme diminished tumourigenesis [246]. On the contrary, as summarised in Table 1, the enzyme 11-β HSD-2 was undetected in OSCC [18].

### 3.6. Role of Stress Hormones in Immunity

Immune dysregulation as a result of psychological stress has earned significant attention [247]. The prolonged activation of neuroendocrine pathways, due to chronic stress affect the immune system and aid in the survival and growth of tumour cells. The immune cells B lymphocytes, T lymphocytes and NK cells have a vital role in providing immunity against tumours. They regulate the immune responses, indirectly, by releasing cytokines and antibodies that recruit other immune cells or by directly killing the cancer cells [87,248,249,250]. Activation of SNS and HPA suppresses the immune functions of T cells [86,251,252,253].

Altering the actions of immune cells is one of the primary ways in which chronic stress promotes cancer development [87]. The early evidence for the role of chronic stress comes from studies that observed that increased cortisol levels reduced the number of lymphocytes [254,255,256,257,258].

Both B and T lymphocytes have a vital role in immune responses against cancer. They either directly kill cancer cells or produce cytokines and antibodies that assist in recruiting other immune cells to do so. Studies in mice have shown a correlation between stress and decreased T cell mediated immunosurveillance in tumours [259,260]. Elevated β-adrenergic signalling resulted in suppressed proliferation and cytolytic properties of CD8 + T cells and caused lymphoma progression in a mouse model [261]. A recent study found that a lower increase in GR sensitivity of the immune cells was associated with increased fatigue in patients with head and cancer [262].

## 4. Stress Measurement

The complex aetiology of stress and its varied reactions in different people, makes it difficult to find a gold standard for stress measurement. Studies so far have used stress assessment scales that employ questionnaires, which are a subjective tool to report measures of wellbeing or illbeing [263,264]. These include Perceived Stress Scale (PSS) [265], Depression Anxiety and Stress Scale (DSS) [266], Ryff Psychological Wellbeing Scales [267], and Satisfaction with Life Scale (SwLS) [268]. On the other hand, the physiological measures of stress use urine and blood samples to measures the levels of cortisol and other stress hormones [227]. Salivary samples have also been used to give a measure of cortisol [18]. However, in recent years, the traditional samples of blood, urine and saliva are thought to provide only an acute measure of cortisol as opposed to the long-term exposure and therefore analysis of hair cortisol concentration (HCC), has appeared as an important measurement of the biomarker cortisol, which allows the investigation of a long-term cortisol exposure. It is also non-invasive and less burdensome for the patients, in contrast to urine, blood and saliva [269,270]. Studies have found inconclusive results when assessing the relationship between stress questionnaire responses and HCC. For some studies a positive association is seen whereas for others, a negative association is observed. This is explained by the recall bias associated with the questionnaires [271]. Many studies have used HCC as a measure of stress in war hit areas, pregnancies, academic settings and work environments [272,273,274,275,276,277]. However, cancer studies have not used this method so far.

## 5. Conclusions

The phrase “Healthy mind in a healthy body” comes from archaic times when the ancient Greeks understood the need for harmony between the mind, body and spirit in order to maintain a good physical and mental health. Stress is an integral part of our lives and becomes inescapable with the diagnosis of a serious illness, such as cancer. Over the last two decades, the activity in science has risen to understand the effect of stress on initiation and progression of cancer. Whereas some forms of cancer such as breast, prostate, lung and ovarian have seen the role of stress hormones more thoroughly characterised, oral cancer still lags behind despite being amongst the most stressful cancers [278,279].

The challenges faced by the HNC patients support the need for increased attention and more elaborate research on the mechanisms through which stress can progress the disease. When it comes to the aetiology of HNC, factors such as alcohol, tobacco and viral infections remain prominent. The role of stress in progression is considered only to be limited to encourage behaviours that lead to increased alcohol and smoking whereas the evidence reviewed here shows that stress hormones, along with dampening the activity of immune system, directly affect all facets of the hallmarks of cancer. This makes it paramount for future studies to carry out more in-depth research on stress signalling in HNC. The challenge this area of research presents is in the understanding and interpretation of stress, since the same stressful experience can generate a different stress response in different people. However, this makes it even more important to understand the psychological health of cancer patients, as the same level of disease may be progressing at different rates due to different stress levels.

To date, in attempting to determine the relationship between stress and cancer, many studies have used stress-based questionnaires and blood or salivary samples to measure stress hormones. The challenge with blood and salivary samples is that they provide only a short-term measure of stress. Recent stress studies have utilised hair to give a long-term measure of the stress hormone cortisol. However, cancer studies have not made use of this yet. Part of our group’s studies will be to utilise our cohort of patients in terms of questionnaires, cortisol level in hair, biopsy and cell culture studies to investigate this.

Large scale further research studies in HNC are required to better understand and measure stress and also to intensify research on the underlying molecular mechanisms through which stress influences HNC. This would help understand the relationship between stress and cancer, which could be used to minimise the spread of the disease. The incorporation of a multidisciplinary rehabilitation, with a psychological intervention may aid patients in the significant challenges they face associated with HNC, may improve Quality of Life and serve to slow down progression of disease.

## Figures and Tables

**Figure 1 cancers-13-00163-f001:**
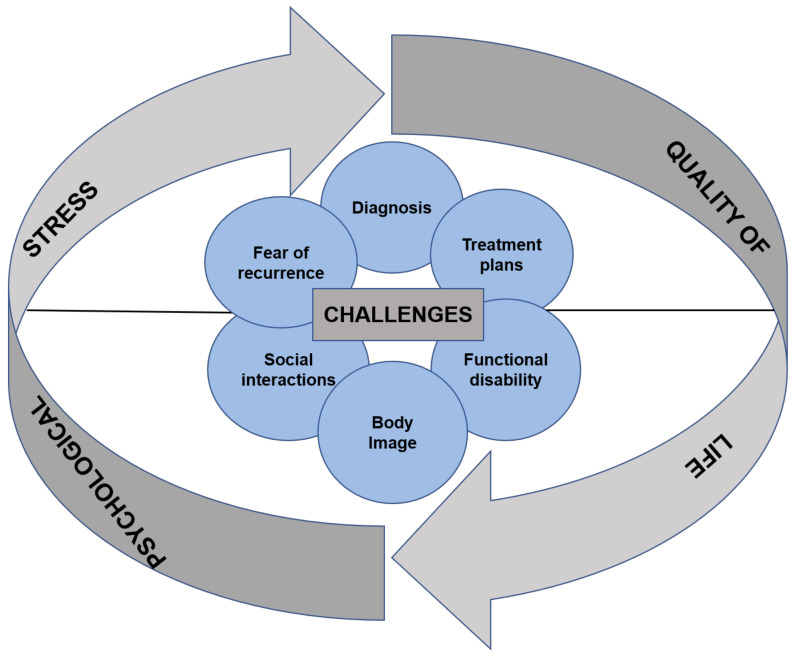
Summary of the challenges associated with head and neck cancers. The diagnosis itself, complex treatment plans, functional disability in terms of speech and mastication, concerns around body image that in turn lead to compromised social interactions, and fear of recurrence (FCR) are some of the challenges faced by the patients with head and neck cancers (HNC). These challenges decrease the quality of life (QoL) and increase psychological stress.

**Figure 2 cancers-13-00163-f002:**
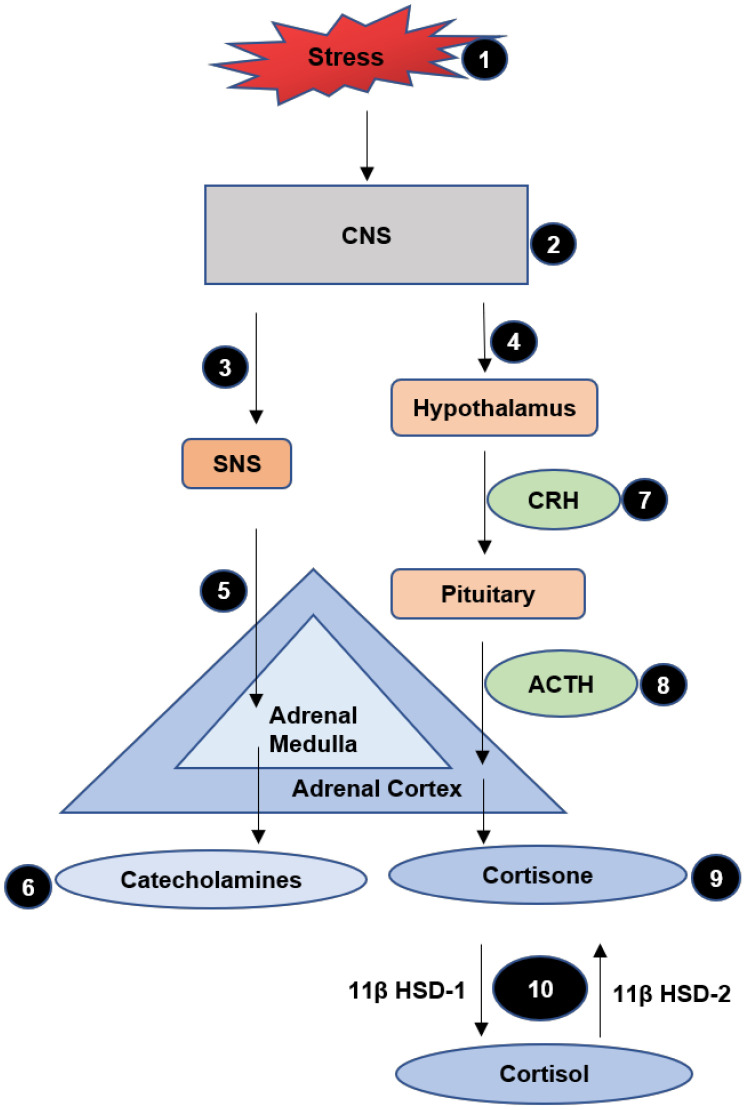
Response of the body to stress. A stressor (**1**), causes the central nervous system (CNS) (**2**), to activate the sympathetic nervous system (SNS) (**3**), and hypothalamus–pituitary–adrenal axis (HPA) (**4**). Sympathetic nervous system (SNS) activates adrenal medulla (**5**), which releases catecholamines (**6**). Hypothalamus releases corticotropin-releasing hormone (CRH) (**7**), which causes the pituitary gland to release adrenocorticotropic hormone (ACTH) (**8**). Adrenocorticotropic hormone (ACTH) results in the release of cortisone (**9**), from adrenal cortex. Cortisone is activated by the enzyme 11-β hydroxysteroid dehydrogenase type 1 (11-β HSD-1) to cortisol and 11-β hydroxysteroid dehydrogenase type 2 (11-β HSD-2) to cortisone, in the target organs (**10**).

**Figure 3 cancers-13-00163-f003:**
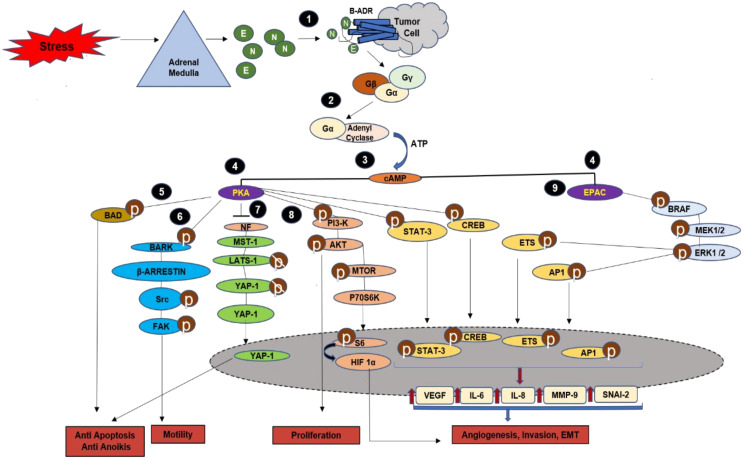
Adrenergic signalling pathway and mechanisms. The binding of Epinephrine (E) and Norepinephrine (N) to β-adrenergic receptors (β-ADR) results in Gαs-mediated activation of adenylyl cyclase (**1**,**2**). This causes a transient influx of cyclic AMP (cAMP) (**3**). cAMP activates the two effector pathways, Protein Kinase A (PKA) and exchange protein directly activated by cyclic AMP (EPAC) (**4**). PKA phosphorylates Bcl-2 associated agonist of cell death (BAD) which makes the cells resistant to apoptosis and anoikis (**5**). It also phosphorylates β-adrenergic receptor kinase (BARK) which recruits β-arrestin, further phosphorylating Src and Focal Adhesion Kinase (FAK), resulting in cell motility (**6**). In cervical cancer cells, sustained adrenergic signalling that results in PKA activation causes inhibition of the tumour suppressive Hippo Yap pathway. PKA targets the tumour suppressor Neurofibromin 2 (NF-2), as a result of which the downstream phosphorylation of mammalian Ste20-like kinases ½ (MST1/2; homologs of Drososphila Hippo (Hpo)), large tumour suppressor ½ (LATS ½; homologs of Drosophila Warts (Wts)) and Yes-Associated Protein (YAP) is inhibited. The dephosphorylated YAP translocates into the nucleus and inhibits apoptosis (**7**). β-adrenergic receptors activated by stress lead to cAMP-PKA/AKT/mTOR/P70S6K/HIFα pathway-dependent proliferation and angiogenesis (**8**). EPAC leads to the activation of BRAF-MAPK signalling pathway (**9**). The transcription factors STAT3, CREB, ETS, AP1 are phosphorylated by PKA as well as by EPAC (**10**) which upregulate the expression of vascular endothelial growth factor (VEGF), interleukin (IL)-6, IL-8, matrix metalloproteinases (MMPs), SNAI-2 involved in angiogenesis and invasion. Adapted from [91,94,96,100].

**Figure 4 cancers-13-00163-f004:**
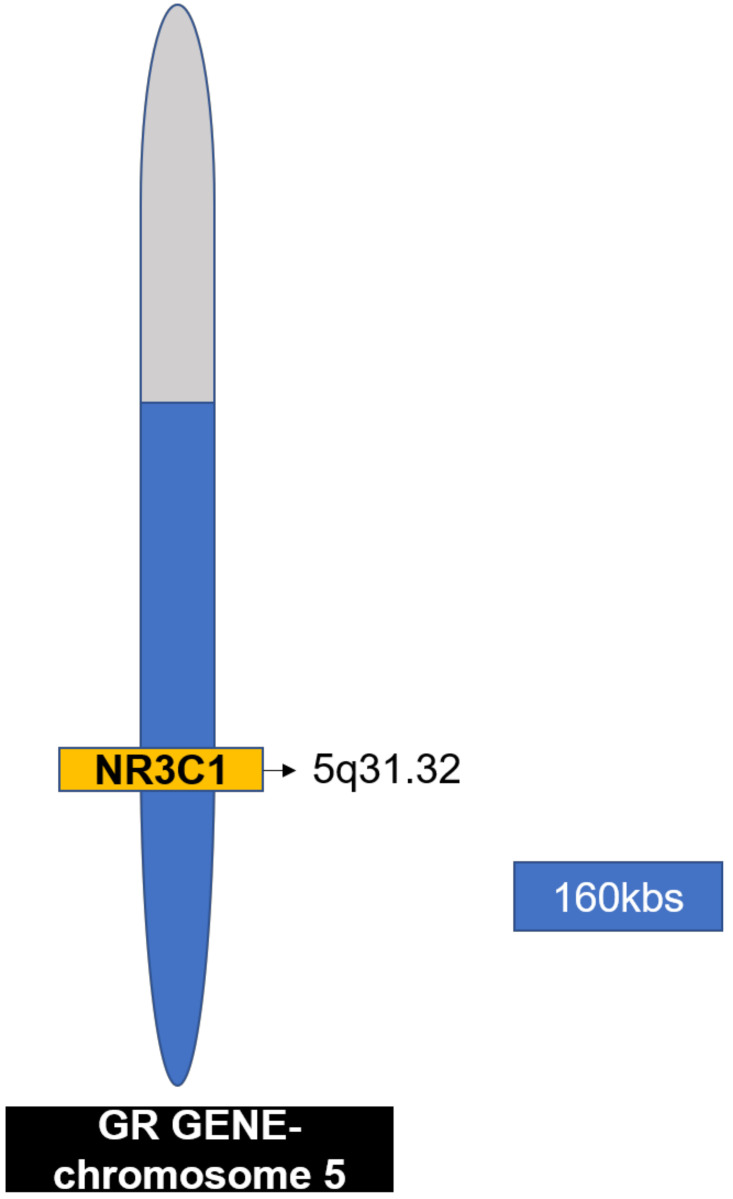
The human glucocorticoid receptor (hGR) gene. The hGR gene, called NR3C1 is located on chromosome 5 (5q31.32) and spans 160kb.

**Figure 5 cancers-13-00163-f005:**
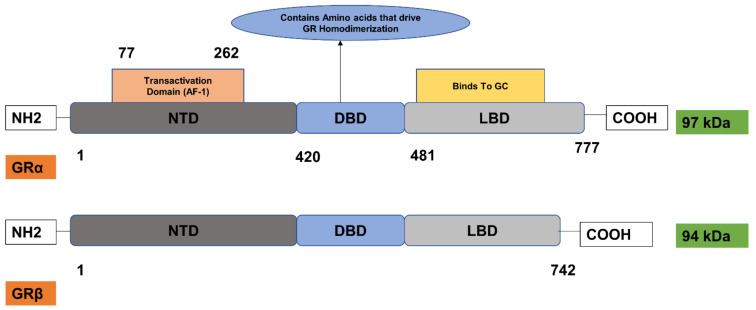
Glucocorticoid receptor α and β isoforms. Alternative splicing of the human glucocorticoid receptor gene (hGR) in exon 9, results in two receptor isoforms GRα and β, which differ at the ends of their C-termini by the number of amino acids in the ligand-binding domain (LBD). The three major domains of the hGR are N-terminal domain (NTD), DNA-binding domain (DBD) and ligand-binding domain (LBD). The N-terminal domain contains a major transactivation domain called Activation Factor-1 (AF-1), critical for the transcriptional activation of the receptor. DNA-binding domain (DBD) contains amino acids responsible for GR homodimerisation. The LBD consists of the ligand-binding site. GRα contains 50 additional amino acids in its ligand-binding domain (LBD) with a molecular weight of 97 kDa and GRβ contains 15 amino acids with a molecular weight of 94 kDa. Of the two isoforms, only GRα binds to the glucocorticoids. Adapted from [109].

**Figure 6 cancers-13-00163-f006:**
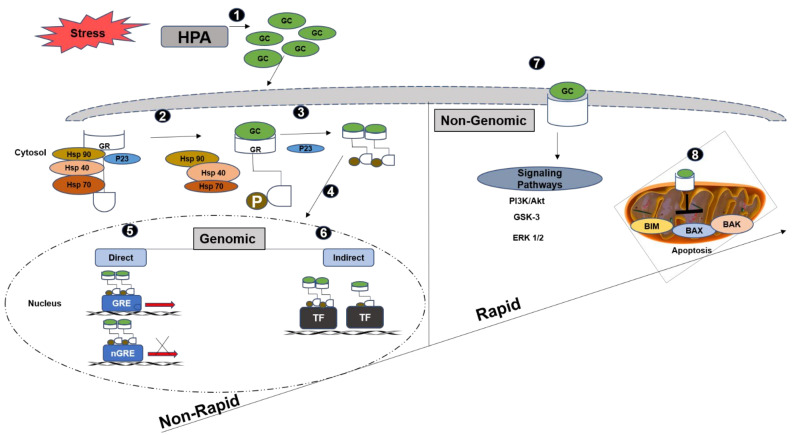
Genomic and non-genomic glucocorticoid signalling. Hypothalamic–pituitary–adrenal axis (HPA) releases glucocorticoids (GC) in response to stress (**1**). GC binding to cytosolic glucocorticoid receptor (GR) results in the dissociation of heat shock proteins (Hsp 90, Hsp 40, Hsp 70) and brings a conformational change leading to the phosphorylation and dimerisation of GR. (**2**,**3**). The GC-GR complex translocates into the nucleus (**4**), where it may result in transcriptional upregulation by direct DNA binding to positive glucocorticoid response elements (GRE) and down-regulation by binding to negative glucocorticoid response elements (nGRE) (**5**) or indirect DNA binding via transcriptional factors (TF) (**6**). The rapidly occurring non genomic mechanisms take place through the membranous glucocorticoid receptor (mGR) (**7**) that activate the proliferative signalling pathways, and cause inhibition of apoptosis by translocation of GR to mitochondria (**8**). Adapted from [125].

**Table 1 cancers-13-00163-t001:** Studies on Head and Neck Cancer Showing the Effect of Stress Hormones and Receptors.

Study	Performed on	Techniques Used	Findings
1. Stress hormones increase cell proliferation and regulates interleukin-6 secretion in human oral squamous cell carcinoma cells [9].	Oral squamous cell carcinoma (OSCC) cell lines (SCC9, SCC15, SCC25), 20 OSCC biopsies, 17 leukoplakia biopsies, 15 Normal oral mucosae.	Polymerase Chain Reaction (PCR), to determine Interleukin-6 (IL-6) gene expression in cell lines and tissues, (3-(4,5-dimethylthiazol-2-yl)-2,5-diphenyltetrazolium bromide) (MTT) to determine cell proliferation, enzyme-linked immunosorbent assay (ELISA) to determine the IL-6 protein levels.	Norepinephrine (NE) increased IL-6 expression and cell proliferation in OSCC cell lines. Pharmacological dose of cortisol decreased VEGF and IL-6 whereas, stress dose increased VEGF and IL-6 expressions. Mean expression of β1 mRNA in OSCC was higher compared to normal mucosa (*p* < 0.05).
2. Chronic stress promotes oral cancer growth and angiogenesis with increased circulating catecholamine and glucocorticoid levels in a mouse model [10].	Oral Cancer Cell line CAL 27 implanted into mice.	Catecholamine levels were determined by High Performance Liquid Chromatography with Mass spectrometry (HPLC-MS/MS). Expression of VEGF and MMP by was observed with immunohistochemistry (IHC). Physical restraint system was used to induce characteristic chronic stress.	Chronic stress increased tumour size, matrix metalloproteinases (MMP), VEGF expression, level of plasma catecholamines, cortisone and caused more invasive growth of oral carcinoma cells in a mice model.
3. Association of Increased Circulating Catecholamine and Glucocorticoid Levels with Risk of Psychological Problems in Oral Neoplasm Patients [11].	75 patients (49 men and 26 women) with oral tumours were included.	Checklist 90-revised Inventory (SCL90-R) which is a self-assessment survey of 90 questions, as well as a demographic questionnaire were used to assess the psychosocial status of patients. Blood samples were taken two hours before surgery, between approximately 9:00 and 11:00 a.m. Catecholamine levels were determined by High Performance Liquid Chromatography with Mass spectrometry (HPLC-MS-MS).	Significant difference in the scores of SCL90-R between the benign tumour and cancer patients’ groups was only seen in the dimensions of depression (*p* = 0.0201) and obsessive-compulsion (*p* = 0.0093) Peripheral blood mean concentrations of catecholamines and glucocorticoids in the oral cancer group were higher than in the benign tumour group (*p* < 0.01) (*p* < 0.001), respectively. Stage I and II cancer showed comparatively low concentrations of epinephrine and Stage III and IV cancer showed substantially greater concentrations of epinephrine, norepinephrine, cortisone, hydrocortisone.
4. Prognostic significance of beta-2 adrenergic receptor in oral squamous cell carcinoma [12].	Clinicopathological data, treatment, tumour outcome, prognosis and expression of β2-adrenergic receptor was examined for 106 OSCC patients.	Immunohistochemistry was used to analyse the expression of β2-adrenergic receptors and its relation to clinicopathological variables.	Strong cytoplasmic and membranous β2-adrenergic receptor expression was found in malignant OSCC (72.6%). Significant association between β2-adrenergic receptor expression and alcohol (*p* = 0.021), simultaneous use of alcohol and tobacco (*p* = 0.014) and T stage (*p* = 0.07), was observed.
5. Expression of β2-adrenergic receptor in oral squamous cell carcinoma [13].	65 OSCC patients with pathologically confirmed diagnosis of OSCC. Ten cases of adjacent normal mucosa as controls. TCa8113—cell line from OSCC of tongue. ACC—cell line from Salivary Adenoid Cystic Carcinoma.	Immunohistochemistry (IHC) Western blot RT PCR Migration assay Proliferation assay.	β2 expression significantly correlated with cervical lymph node metastasis (*p* = 0.001), age (0.003), tumour size (0.001), clinical stage (0.001).
6. Glucocorticoids reduce chemotherapeutic effectiveness on OSCC cells via glucose-dependent mechanisms [14].	Oral malignant keratinocytes: H314, H357, H400, BICR16, BICR56.	Annexin V-FITC assay to study apoptosis. Enzyme-linked immunosorbent assay (ELISA)—to measure the concentration of cortisol after adrenocorticotropic hormone (ACTH) stimulation.	Glucocorticoids had an antiapoptotic and protective effect on OSCC against chemotherapy in a glucose dependent manner.
7. Immunoexpression of glucocorticoid receptor alpha (GRα) isoform and apoptotic proteins (Bcl-2 and Bax) in actinic cheilitis and lower lip squamous cell carcinoma [15].	22 cases of actinic cheilitis (AC), 44 cases of lower lip squamous cell carcinoma (LLSCC) (22 with normal mucosa, 22 without normal mucosa) The percentages of nuclear (GRα) and cytoplasmic (GRα, Bcl-2, and Bax) staining in epithelial cells were correlated with clinical (tumour size/extent and clinical stage) and histopathological parameters (risk of malignant transformation for AC and histopathological grade of malignancy for LLSCCs).	Immunohistochemistry (IHC).	A relatively high median percentages of GRα positive staining was observed in all cases. A lower nuclear expression and higher cytoplasmic expression of GRα was observed in LLSCC specimens compared to actinic cheilitis (*p* < 0.05). A higher GRα expression was observed in high grade tumours compared to low grade tumours (*p* = 0.036).
8. Circulating catecholamines are associated with biobehavioural factors and anxiety syptoms in head and neck cancer patients [16].	Plasma epinephrine and norepinephrine were measured. Psychological anxiety levels in 93 patients with HNSCC and 32 patients with oral leukoplakia.	Plasma epinephrine and norepinephrine were measured by High Performance Liquid Chromatography-Electrochemical Detection (HPLC-ED). Psychological anxiety levels measured by Beck Anxiety Inventory (BAI).	Significantly higher levels of plasma epinephrine and norepinephrine were observed in OSCC patients than in oropharyngeal and oral leukoplakia patients. The total BAI mean scores did not show a significant difference among the three groups.
9. Characterisation of a Novel Oral Glucocorticoid System and Its Possible Role in Disease [17].	Normal oral keratinocytes (NOK), normal oral fibroblasts (NOF), normal oral mucosa (NOM) and malignant tissue.	Western Blot, Immunohistochemistry, ELISA.	NOK and NOF synthesise cortisol in the presence of 10 nM ACTH. NOK expressed 11-β hydroxysteroid dehydrogenase type 1 (11-β HSD1), 11-β hydroxysteroid dehydrogenase type 2 (11-β HSD 2), Glucocorticoid Receptor and Mineralocorticoid Receptor. NOK lacked 11-β-HSD 2 showing their inability to degrade cortisol. 11-β-HSD expression was not detected in OSCC.
10. Increased plasma and salivary cortisol levels in patients with oral cancer and their association with clinical stage [18].	34 oral squamous cell carcinoma (OSCC) patients, 17 oropharyngeal SCC patients, 17 oral leukoplakia patients, 27 smokers and/or drinkers and 25 healthy volunteers.	The plasma and salivary cortisol levels of patients with OSCC were compared with other groups by enzyme immunoassay with a commercial kit.	OSCC patients showed significantly higher levels of plasma (*p* < 0.05) and salivary (*p* < 0.01) cortisol compared to all the other groups. Patients at advanced stage of OSCC showed significantly higher cortisol levels than those at initial stage.
11. The stress hormone Norepinephrine promotes tumour progression through β2-adrenoreceptors in oral cancer [19].	40 OSCC samples from patients and 20 para cancer Normal Oral Mucosa. SCC25 and CAL 27 cell lines.	RT-PCR Immunohistochemistry Cell proliferation assay CCK-8 Matrigel coated transwell assay Colony forming assay Sphere forming assay.	OSCC showed significantly higher β2 adrenergic receptor expression than normal para cancer tissue. Norepinephrine promoted proliferation, invasion and stem cell characteristics of OSCC cell lines.
12. Stress hormones concentrations in the normal microenvironment predict risk for chemically induced cancer in rats [20].	Male Wistar rats	Measurement of stress hormone levels, norepinephrine, corticosterone, ACTH and brain-derived neurotropic factor (BDNF) in the tongue microenvironment prior to carcinogen induction was done by ELISA and Milliplex Multi-Analyte Profiling method. To induce tumours, mice were treated with 4-nitroquinoline-1-oxide. The tongues with carcinogen induced lesions were used to perform histochemical analysis and RT PCR.	Increased concentrations of norepinephrine and BDNF positively correlated to OSCC occurrence whereas decreased basal corticosterone levels were predictive for OSCC occurrence.
13. Activation of adrenergic receptor β2 promotes tumour progression and epithelial mesenchymal transition in tongue squamous cell carcinoma (2018) [21].	Tongue squamous cell carcinoma (TSCC) specimens (*n* = 70) and adjacent non-cancerous tissue samples (*n* = 20). CAL 27 and SCC 15 cell lines.	Immunohistochemistry Cell migration and invasion assay Immunofluorescence.	Increased expression of β-adrenergic receptor was observed in TSCC and was associated with lymph node metastasis and reduced overall survival. Treatment of cells with isoproterenol induced epithelial–mesenchymal transition (EMT) by activating IL-6/STAT-3 SNAIL 1 pathway.

## Data Availability

No new data were created or analyzed in this study. Data sharing is not applicable to this article.

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
