# Peer review of "Cancer and Stress: Does It Make a Difference to the Patient When These Two Challenges Collide?"

_cancers, 2021, doi:10.3390/cancers13020163_

Round 1

Reviewer 1 Report

The work of Iftikhar is very well written and explained in a clear and in depth way the topic. I really appreciate this work.

There are  minor points that have to be elucidated before pubblication 

the figures are too small and not well described-

 the figures shoud be enlarged  in the figure legendshoud be  expaded in  the explanationof pathways

 the new emerging role of the beta 3 receptor in cancer  shoud be addressed in the point 3.

Author Response

 The authors appreciate the kind words of the reviewer and have hopefully answered the points below,

All figures enlarged.

Figure 1- Figure description added to lines, 124- 128

Figure 2 -Figure description edited; abbreviations expanded to full form-(lines 163-171). The small black circles with numbers to explain the steps in the figure changed to improve understanding.

Figure 3- Figure description edited, abbreviations expanded to full forms- lines (208-223).

Figure 4- Figure made simpler, boxes removed.

Figure 5- Figure made simpler- lower summary box removed. Figure description added. (lines, 323-331)

Figure 6- Figure description edited. Abbreviations expanded to full form. (Lines, 333-341).

Beta 3 Adrenergic receptors role in proliferation and angiogenesis in melanoma and neuroblastoma added. Line 389 (Ref 146), Line 437- 488 (ref 172), line 439 (ref 173).

Reviewer 2 Report

The review addresses a very important topic. The text is well written and describes earlier studies written on stress pathways and HNC and other diagnoses. The number of references is impressive. However, it is difficult in the introduction to understand the aim of the study. You have to look at the “simple summary” or "abstract” to understand the aim. The text is written as a long Introduction and ends with a conclusion. I miss headlines as aim, methods, results, and discussion. It would clarify the authors’ great work.

You should be able to read a figure without reading the text of a manuscript, all abbreviations should be explained in the figure text, and that also includes well known abbreviations. Throughout the manuscript, the figure texts need to be improved. E.g. in Figure 2, there is a number of abbreviations such as CNS, SNS, HPA, CRH, and ACTH, which should be written out in the text below the figure. Additionally, in Figure 2 there are two abbreviations in the figure that is not written out in the figure text below. They should be written in brackets and that regards the text 11-β hydroxysteroid dehydrogenase type 1 to cortisone (11-β HSD-1) and 11-β hydroxysteroid dehydrogenase type 2 to cortisone (11-β HSD-1).

Throughout the manuscript, there are extra spaces in the text e.g. lines 82-83 and 288-290. There are many abbreviations throughout the text which makes it a bit unpleasant to read. It would be better if some of the abbreviations that are not necessary are removed.

In Table 1 there is an extra column that lacks text and at the end there is an extra row. I suggest the authors remove the extra column and row.

Author Response

The authors appreciate the positive comments made by the reviewer, We have hopefully answered the points raised in the text below.

Aim/scope of study added to (Lines 48-52)

Headlines aims, method, discussion- This review followed the suggested format of the journal and the example review, “the role of carcinogenesis related biomarkers in the Wnt pathway and their effects on EMT in Oral Cancer.”

Figure text description improved. Abbreviations expanded within the figure description below the figures for figure 2 and the rest of the figures.

The journal outline suggest use of abbreviations where necessary. If the referee/editorial team can advise on the removal of specific ones we would appreciate that. 

We have checked the spacing and the odd formatting in one part of the document and changed as requested.

Extra column and row removed from the table.

Reviewer 3 Report

Dear Authors

The article "Cancer and Stress: does it make difference to the patient when these two challenges collide" is a challenge.

Base on the literature the authors try to establish, understand and explain the effect of stress pathways on Head Neck patients and their effect on the course of a cancer.

This article is a review. But it is necessary to present methods of this review. What databases were taken into account?  What methods were used to select the studies? It needs to be completed.

Author Response

The authors appreciate the positive comments from the reviewer. The point the reviewer made around our method and database search criteria were fairly old fashioned and simple. We used the old-fashioned approach of Pubmed search using the words Head+Neck+Cancer+stress. Results were selected from paper/studies that were open access and in English.

Reviewer 4 Report

This review discussed and summarized the recent studies about the effects of stress on head and neck cancer (HNC). The authors talked about signaling pathways mediated stress response, and summarized how stress hormones affects cancer (including HNC), such as affecting cell proliferation, angiogenesis, cell survival, DNA damage, and microenvironment. One question is that majority of the functional studies of stress and cancer are performed using cancer models other than HNC, why the authors want to focus on HNC? But overall, this is a well written and well organized review paper. I only have some minor points that need to be addressed to improve the paper.

  1. Table 1 is too long. Not necessary to put every conclusions in the paper. Should be concise.
  2. The authors should add more discussion to talk about the future directions.
  3. The authors should pay attention to the writing, especially the grammatical errors and unnecessary spaces.

Author Response

The authors very much appreciate the positive comments given to us by the reviewer. We have addressed the concerns they raised by making the Table more concise. The major question was why focus on HNC? HNC are amongst the top most when it comes to psychological stress due to the challenges highlighted in the first part of the review yet studies are very few compared to other cancers. This is why we present studies done on other cancers, in the review and highlight the need to elaborate research on the role and effect of stress in HNC. We have emphasised that in our future work we will be drawing on the studies from other cancers to be part of our study.

Round 2

Reviewer 2 Report

Great work with the changes.

Reviewer 3 Report

The Authors presented a review method. It should be included in the text of the manuscript.